# Characterization of microbiota and histology of cultured sea cucumber *Isostichopus badionotus* juveniles during an outbreak of skin ulceration syndrome

Karen A. Arjona-Cambranes[1], Miguel A. Olvera-Novoa[1], Daniel Cerqueda-García[2], Madeleine G. Arjona-Torres[3], M. Leopoldina Aguirre-Macedo[1], Víctor M. Vidal-Martínez[1]*, José Q. García-Maldonado[1]*

1 Departamento de Recursos del Mar, Centro de Investigación y de Estudios Avanzados del Instituto Politécnico Nacional (CINVESTAV) Unidad Mérida, Mérida, Yucatán, México, 2 Red de Manejo Biorracional de Plagas y Vectores, Clúster Científico y Tecnológico Biomimic®, Instituto de Ecología, A.C., Xalapa, México, 3 Laboratorio de Patología, Campus de Ciencias Biológicas y Agropecuarias, Universidad Autónoma de Yucatán, Mérida, México

* vvidal@cinvestav.mx (VMVM); jose.garcia@cinvestav.mx (JQGM)

## Abstract

Due to the dramatic reduction of sea cucumber *Isostichopus badionotus* populations in the Yucatan Peninsula by overfishing and poaching, aquaculture has been encouraged as an alternative to commercial catching and restoring wild populations. However, the scarcity of broodstock, the emergence of a new disease in the auricularia larvae stage, and the development of skin ulceration syndrome (SUS) in the culture have limited aquaculture development. This study presents the changes in the intestine and skin microbiota observed in early and advanced stages of SUS disease in cultured juvenile *I. badionotus* obtained during an outbreak in experimental culture through 16S rRNA gene sequencing and histological evidence. Our results showed inflammation in the intestines of juveniles at both stages of SUS. However, more severe tissue damage and the presence of bacterial clusters were detected only in the advanced stages of SUS. Differences in the composition and structure of the intestinal and skin bacterial community from early and advanced stages of SUS were detected, with more evident changes in the intestinal microbial communities. These findings suggest that SUS was not induced by a single pathogenic bacterium. Nevertheless, a decrease in the abundance of *Vibrio* and an increase in *Halarcobacter* (syn. *Arcobacter*) was observed, suggesting that these two bacterial groups could be keystone genera involved in SUS disease.

## Introduction

Sea cucumber *Isostichopus badionotus* is the holothuroid with the most significant commercial interest in the southeastern region of the Gulf of Mexico and the Great Caribbean due to its high demand and acceptance in the Asian market, where it can reach prices between 132 and 358 USD Kg$^{-1}$ [1, 2]. However, the intensive fishing of this species since 2010, as well as the

**Data Availability Statement:** All relevant data are within the manuscript and its Supporting Information files.

**Funding:** This research was supported by the Mexican National Council for Humanities, sciences and Technologies- CONAHCYT – Mexican Ministry of Energy- Hydrocarbon Fund, project 201441; Fomix-CONAHCYT project No.246841. CONAHCYT awarded KAAC with doctoral scholarship No. 775910. This is a contribution of the Gulf of Mexico Research Consortium (CIGoM). The funders had no role in study design, data collection and analysis, decision to publish, or preparation of the manuscript.

**Competing interests:** The authors have declared that no competing interests exist.

illegal fishing that continues occurring, have resulted in the overexploitation of wild populations [2–5]. Furthermore, the densities of this species do not recover quickly after several years of no fishing [4].

The development of the experimental culture of *I. badionotus* in Yucatan, Mexico, started more than a decade ago at our research institute (Cinvestav). However, the research efforts have primarily focused on aquaculture technology, nutrition, and feeding, as well as its nutraceutical or pharmacological potential [6–11]. Recently, this species' gut microbiota composition in captivity was reported using high-throughput 16S rRNA sequencing [12].

Skin Ulceration Syndrome (SUS) or Skin Ulceration Disease (SKUD) is a common disease reported in cultivated sea cucumbers worldwide. Previous studies have shown that it can be induced by microbial (bacteria or viruses) or abiotic factors (high temperature, salinity, and stocking densities) [13]. SUS is a highly infectious and lethal disease of sea cucumbers that starts with white spots on the skin, followed by deep and extended ulcers that expose the underlying muscles and spicules [14, 15].

The study of this disease has previously focused on isolating bacteria from ulcers. Nevertheless, some studies have performed experimental infection tests with the isolated bacteria without success in inducing SUS [13, 16]. It is known that environmental stressors generate disorders in the symbiotic microbial community, causing diseases in holothuroids such as *Apostichopus japonicus* [17, 18]. In recent years, there have been reports of a link between variations in the intestinal microbiota and the development of SUS in sea cucumbers [19–21]. However, few studies have used histology to describe tissue damage caused by the disease [13, 21].

An outbreak in the experimental culture of sea cucumbers *I. badionotus* occurred in Yucatan in 2018. Due to the rapid spread of the disease and the limited number of organisms that survived in culture conditions, only two juveniles with no lesions on the skin surface and two juveniles with evident ulcers were collected. New juveniles could not be obtained due to a new disease that caused 100% mortality in the auricularia larval stage. This situation makes the limited *I. badionotus* juvenile samples from the 2018 SUS outbreak relevant to identifying the causative agent of the disease to try to prevent its presence. However, the lack of information on the microbial communities of *I. badionotus* has limited the advance in the knowledge of this disease and the development of treatments. Thus, the aims of the present study were: 1) to characterize the intestinal and skin microbiota of early and advanced stages of SUS on *I. badionotus* juveniles cultured in Yucatan, Mexico, and 2) to determine the type and extension of tissue damage by histology.

## Materials and methods

### Juvenile culture system

All *I. badionotus* juveniles were maintained in 6 L polycarbonate containers (17 × 17 × 18.5 cm) filled with natural seawater from a 24-m deep beach well. The seawater was filtered using a sand filter and treated with ultraviolet (UV) radiation. Furthermore, the sand substrate at the bottom of the containers was also subjected to UV radiation treatment. The tanks were part of a closed recirculating system formed by a sand filter, a biological filter, continuous UV radiation (2 x 40 W), and a water chiller fixed at 25°C. The water temperature was maintained at 24 ± 1°C, with a salinity of 35 ±1 0/00 and continuous aeration. To remove faces and unconsumed food, 60% of the seawater was replaced every two days. Juveniles were fed every 2 days with 1 g of a mixture (1:1:1) of *Spirulina* sp. (Bio-Marine Spirulina, Aquafauna Bio-Marine Hawthorne, CA) powder and bakers yeast, reinforced with AlgaMac-2000 (Bio-Marine Inc. Hawthorne, CA), and completed with 15,000 cell/mL of *Thalassiosira weissflogii* (TWcc) to each container [6, 9].

## Description of SUS outbreak in the culture of *I. badionotus*

In February 2018, an outbreak of SUS of juvenile *I. badionotus* was detected in the aquaculture facilities of Cinvestav's marine station at Telchac Puerto, Yucatan, Mexico. The juveniles began to develop skin ulcers which spread over days, causing the organisms to die as well as spreading the disease if the affected juveniles were not separated from the healthy ones.

This SUS outbreak occurred in the last group of juveniles successfully obtained in the experimental culture. In addition, the emergence of a new disease in the culture producing 100% mortality in the auricularia larval stage, has not permitted the development of new juveniles under culture conditions.

## Sample collection

To characterize the etiology of the SUS outbreak in the experimental culture of *I. badionotus*, four juveniles were collected. Two juveniles were initially considered healthy as they had no skin lesions or signs of disease. However, it was later determined that they were in an early stage of SUS due to pathologies observed in intestinal and skin tissues. The other two juveniles exhibited signs of disease, manifested by visible ulcers on their skin, and were classified as being in the advanced stage of the disease.

The organisms were transported to the laboratory in separated seawater containers, maintained at a temperature of 24˚C, and supplemented with oxygen. Subsequently, the juveniles were subjected to a 24-hour acclimation period in filtered and UV-treated seawater, following the above-mentioned culture conditions. After acclimation, euthanasia was performed using a hypothermia protocol, involving immersion in seawater at 12˚C for 75 minutes [22].

To mitigate potential contamination from waterborne microorganisms, the external surfaces of all four sea cucumber juveniles were washed with sterile distilled water, after which they were dissected as previously described [15]. The entire intestinal tract was aseptically removed from the abdominal cavity and subsequently divided into anterior and posterior segments, with two samples collected from each section per juvenile. For molecular analysis, we collected eight skin samples from juveniles in the early stage of SUS (four samples from each juvenile). Additionally, seven skin ulcer samples were collected from juveniles in the advanced stage of SUS (three ulcer samples from one organism and four from the other one). All collected samples were preserved in DNA/RNA shield buffer and stored at -20˚C until further analysis.

## Histological process

Histological transverse sections from the intestine and body wall of both early and advanced SUS stages in *I. badionotus* juveniles were processed. These tissues were fixed in neutralized formaldehyde for 24 h and processed by routine paraffin embedding technique [23]. Histological sections of 5 μm thick were obtained using a Minot rotary microtome (Kedee; Jinhua, ZJ) and stained with hematoxylin-eosin (H-E) to be observed under a Leica DM2500 microscope (Leica; Wetzlar, DE).

## DNA extraction, 16S rRNA amplification, and sequencing

DNA extraction from intestinal and skin samples at both early and advanced SUS stages in *I. badionotus* juveniles was conducted following the manufacturer's instructions, using the Ultra-Clean® Tissue & Cells DNA Isolation Kit (Mo Bio Laboratories, Inc.). For the amplification of the 16S rRNA gene, a nested PCR method was employed. In the first round of PCR, the primer pair 16S.S (5′-AGAGTTTGATCCTGGCTC-3′) and 16S.R (5′-CGGGAACGTATTCACCG-3′) was utilized, resulting in an approximately 1,400 bp fragment

amplification [24]. Subsequently, in the second round, a 500 bp DNA fragment corresponding to the V3 and V4 regions of the 16S rRNA genes was amplified using an Illumina 16S primer pair [25]. Nested PCR reactions were conducted in a final volume of 20 µL, comprising 10 µL of Phusion Flash High-Fidelity PCR Master Mix (Thermo Scientific; Waltham, MA), 0.5 µL of each primer (10 µM), and 1 µL of DNA. The thermal cycling conditions for the first PCR round included an initial denaturation at 94˚C for 3 min, followed by 28 cycles at 94˚C for 30 s, an annealing temperature of 53˚C for 1 min, and an extension at 72˚C for 1 min, concluding with a final elongation step at 72˚C for 10 min. For the second round of amplification, the condition included an initial denaturation at 95˚C for 3 min, followed by 25 cycles at 95˚C for 30 s, an annealing temperature of 55˚C for 30 s, and an extension at 72˚C for 30 s, ending with a final elongation step at 72˚C for 5 min [25].

Metagenomic sequencing libraries for the 16S rRNA gene were prepared according to the manufacturer's protocol. Sequencing was performed on an Illumina MiSeq platform (Illumina, San Diego, CA, USA) with the MiSeq reagent kit V2 (2 x 250), following the manufacturer's instructions. The datasets generated in this study have been deposited in the NCBI database under the BioProject accession number PRJNA980727.

## Bioinformatic analysis

The demultiplexed paired-end reads in the FASTQ format were processed with the QIIME2 pipeline 2019.1 [26]. Error correction and denoising to resolve the reads of Illumina's Amplicon Sequence Variants (ASVs) were performed using the DADA2 plugin, and chimeras were removed using the "consensus" method [27, 28]. Taxonomic assignments of representative ASVs were carried out with the V-SEARCH consensus taxonomy classifier plugin [29], using the SILVA database (v.132) as the reference.

Abundance data was exported to the R environment and the statistical analysis and visualization were conducted using the phyloseq [30], vegan [31], and ggplot2 [32] libraries. Samples rarefaction was performed to a depth of 10,300 reads per sample.

Statistical analyses of intestinal samples from *I. badionotus* juveniles included both sections of the digestive tract (anterior and posterior). This approach was adopted to allow comparison exclusively between the early and advanced stages of SUS disease. Alpha diversity indices, including Observed ASVs, Shannon, and Simpson diversity indices were generated using the "vegan" package. A Wilcoxon-Mann-Whitney (WMW) test was applied to identify significant differences in alpha diversity indices among juvenile groups in each sample. A significant level (*P*-value) less than 0.05 was considered.

A Beta diversity Permutational Analysis of Variance (PERMANOVA) test with 1,000 permutations with the weighted UniFrac distance was conducted to assess significant differences among the conditions of the juvenile samples. Additionally, a Principal Coordinate Analysis (PCoA) based on phylogenetic information and the weighted UniFrac distance was performed [33].

Finally, the characterization of the intestinal and skin microbiota of *I. badionotus* juveniles in both SUS stages was carried out using the linear discriminant analysis (LDA) effect size (LEFSE) method [34].

## Results

### Histological analysis of intestine and body wall of the early and advanced stages of SUS in *I. badionotus* juveniles

Five tissue layers were identified in the posterior intestine of cultured *I. badionotus* juveniles. These layers consist of a mucosa (luminal epithelium) composed of simple columnar

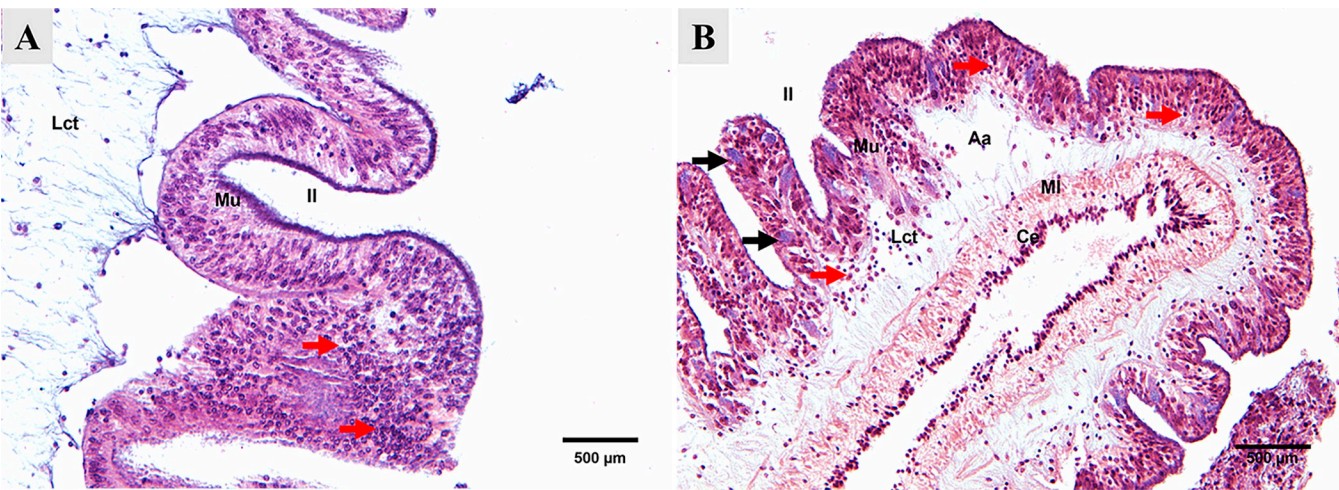

**Fig 1. Histological cross-section of the posterior intestine of cultured *I. badionotus* juveniles.** (A) juvenile I. *badionotus* with SUS in the early stage. H-E. 20X. (B) juvenile I. *badionotus* with SUS in the advanced stage. Round to oval structures compatible with possible bacterial clusters are observed (black arrow). Coelomocyte infiltration (red arrow). H-E. 20X. Il: Intestinal lumen. Mu: mucosa. Lct: Lax connective tissue. Ml: muscular layer. Ce: Coelomic epithelium. Aa: anhistic areas.

epithelium, a submucosa of connective tissue, two muscular layers (longitudinal and circular), and the coelomic epithelium.

In juveniles at the early stage of SUS, most sections of the luminal epithelium displayed typical epithelial cell architecture, although some areas were structurally absent. Notably, in the submucosa, coelomocyte infiltration was observed (Fig 1A). In contrast, advanced stages of SUS in juveniles revealed evident tissue damage. Larger areas without epithelium were observed in the mucosa, and round to oval morphology suggesting possible bacterial clusters within the luminal epithelium, accompanied by submucosal coelomocyte infiltration (Fig 1B). Additionally, anhistic areas, representing regions without tissue, were identified in the connective tissue (Fig 1B).

The composition of the body wall in *I. badionotus* juveniles consisted of five successive tissue layers: the outer cuticle, epidermis, dermis, muscle layer, and coelomic epithelium (Fig 2A and 2B). One of the juveniles with early SUS stage presented some pathologies despite not having damage on the cuticle and epidermis. These pathologies included liquefactive necrosis with anhistic areas in the dermis (Fig 2C), a disordered pattern and degeneration (including the absence of a nucleus in some cells) of muscle fibers, and a loss of coelomic epithelium structure (Fig 2D).

In juveniles with advanced stages of SUS, the principal histological damages included a loss of continuity in the cuticle and epidermis layers, exhibiting varying degrees of severity. Within the dermis, focal areas of liquefactive necrosis with anhistic regions and a disorder in the pattern of connective tissue fibers were observed (Fig 2E). The muscle layer showed a disordered pattern of muscle fibers and signs of degeneration, while the coelomic epithelium exhibited a loss of structure due to disorganized or absent cells in some sections (Fig 2F). Finally, coelomocyte infiltration was consistent in all identified pathologies (Fig 2E and 2F).

## Diversity of intestinal and skin bacterial communities

A total of 608,307 filtered 16S rRNA sequences were obtained from the intestine and skin samples of *I. badionotus* juveniles (293,708 sequences from the intestine and 314,599 from the skin). The sequences were assigned to 683 and 578 ASVs affiliated with bacteria in the intestine and skin samples, respectively (S1 Table). These ASVs were classified into 15 bacterial phyla,

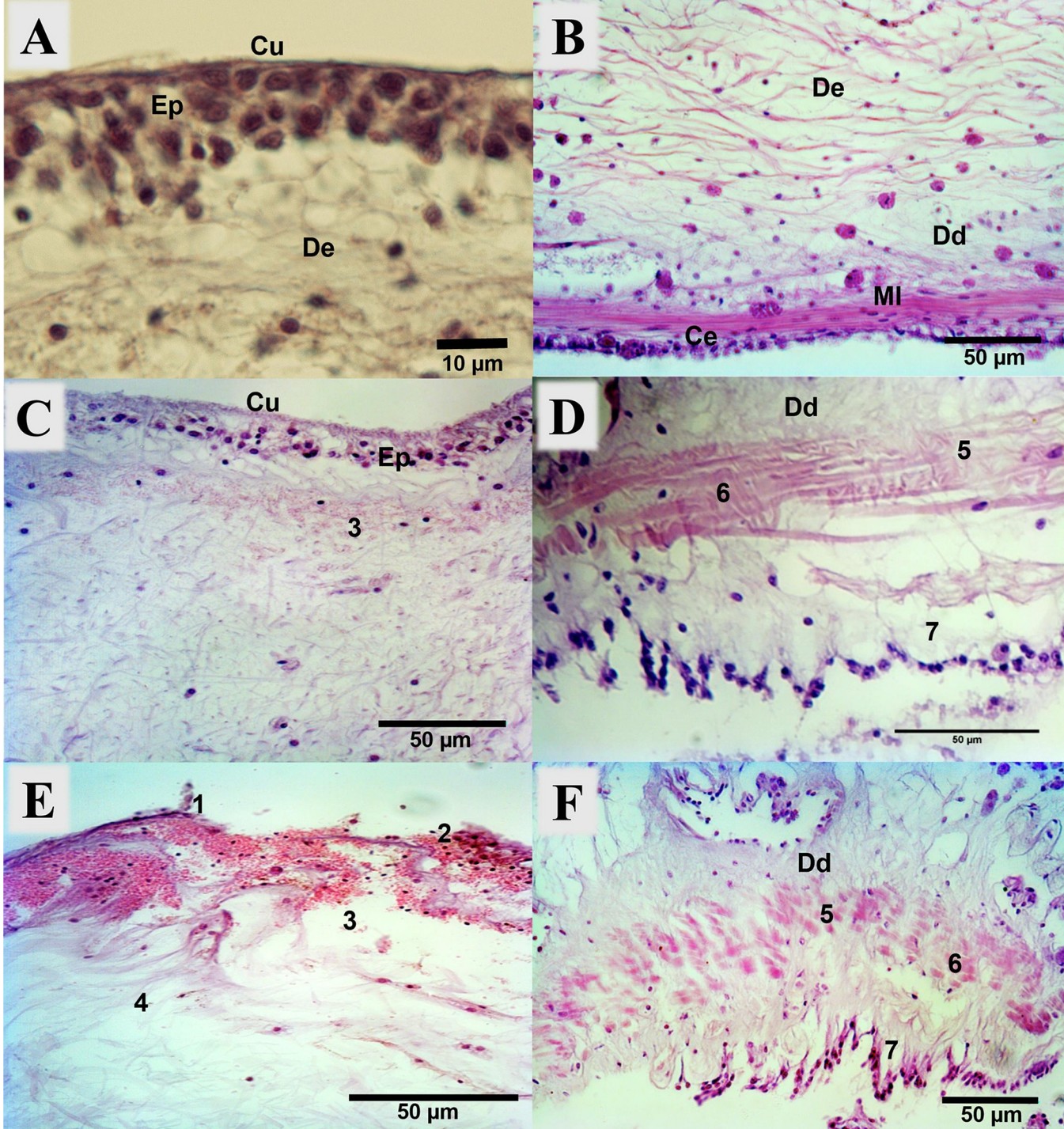

**Fig 2. Histological cross-section of the body wall of *I. badionotus* juvenile.** (A) Healthy body wall layers of an *I. badionotus* juvenile with early SUS stage. H-E. 100X. (B) Healthy body wall layers of an *I. badionotus* juvenile with early SUS stage (continuation). H-E. 40X. (C, D) Body wall layers of an *I. badionotus* juvenile with early SUS stage. H-E. 40X, 60X. (E, F) Body wall layers of an *I. badionotus* juvenile with advanced SUS stage. H-E. 40X. Cu: cuticle; Ep: epidermis; De: dermis; Dd: dense dermis; Ml: muscle layer; Ce: coelomic epithelium 1: Loss in the cuticle continuity; 2: Loss in epidermis continuity; 3: Liquefactive necrosis foci with anhistic areas (areas without tissue) in dermis and coelomocytes presence; 4: Disordered in the pattern of connective tissue fibers with coelomocytes presence: 5: Disordered of the pattern of muscle fibers; 6: Degeneration of muscle fibers; 7: Loss of coelomic epithelium structure.

**Table 1. Richness and alpha diversity indices for bacterial communities in samples of *I. badionotus* juveniles with SUS.**

| Tissue sample | Juvenile SUS stage | Observed ASVs | Shannon diversity index | WMW-test *P*-value | Simpson diversity index | WMW-test *P*-value |
|---|---|---|---|---|---|---|
| Intestine | Early | 91.9 | 2.6±0.6 | 0.960 | 0.8±0.1 | 0.795 |
| | Advanced | 78.4 | 2.6±0.3 | | 0.8±0.1 | |
| Skin | Early | 70.0 | 1.8±0.5 | 0.453 | 0.7±0.1 | 0.222 |
| | Advanced | 90.7 | 2.1±0.5 | | 0.8±0.1 | |

(Mean±SD); ASVs: Amplicon Sequence Variants; WMW: Wilcoxon-Mann-Whitney.

29 classes, 78 orders, 128 families, and 191 genera in the intestine samples and into 14 bacterial phyla, 22 classes, 58 orders, 101 families, and 141 genera in skin samples. The composition of bacterial communities and relative abundance differed according to the early or advanced SUS disease stage of juveniles.

Species richness and alpha diversity indices (S2 Table) for the bacterial communities from the intestine and skin of early and advanced stages of SUS in *I. badionotus* juveniles are presented in Table 1. The species richness for the observed ASVs was higher in the intestine and lower in the skin samples of early-stage SUS juveniles. The Shannon and Simpson indices also revealed higher alpha diversity in the intestinal microbiota than in the skin, but no significant differences were found between the early and advanced stages of SUS in the intestines of juveniles (Table 1). Meanwhile, the diversity of the skin microbiota was higher in juveniles with advanced stages of SUS. However, the WMW test demonstrated no significant differences in the alpha diversity indices in the intestine and skin microbiota between the early and advanced stages of SUS *I. badionotus* juveniles ($P > 0.05$).

A PCoA analysis with weighted UniFrac distance was used to compare the composition of the intestinal and skin microbiota of juveniles (Fig 3). The two axes of PCoA explained 70.6%

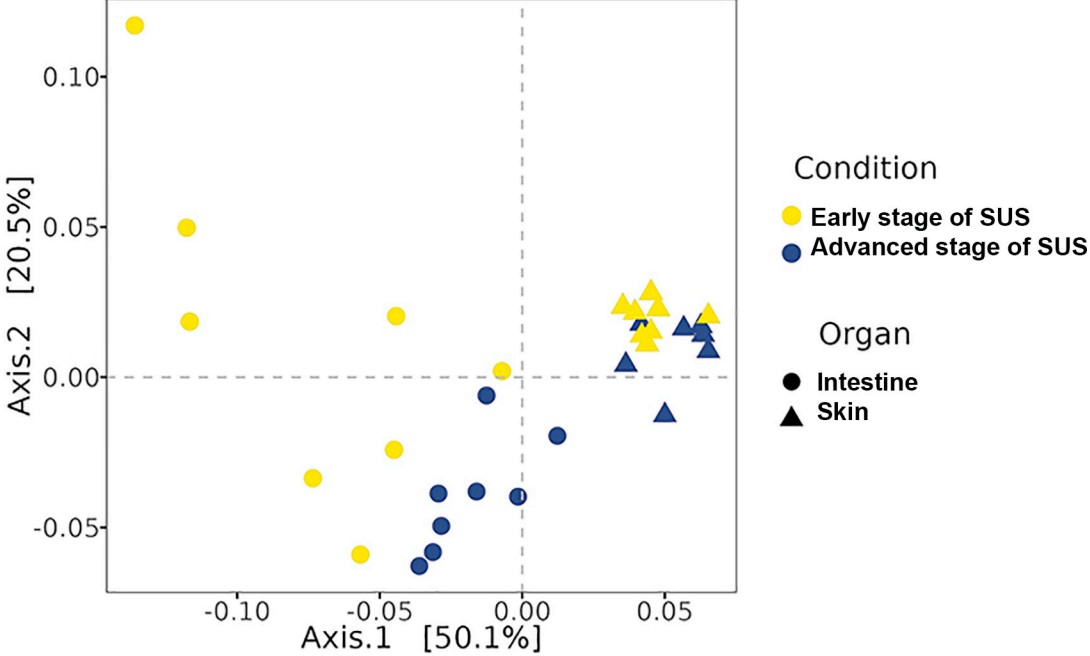

**Fig 3. Principal Coordinate Analyses (PCoA) of bacterial communities in *I. badionotus* juveniles with SUS.** The PCoA analysis was based on a weighted UniFrac distance metric reflecting the beta diversity of the intestinal and skin bacterial communities of cultured *I. badionotus* juveniles in early and advanced stages of SUS.

of the total variance in bacterial structure and illustrated differences between intestinal and skin samples of *I. badionotus* juveniles. The intestinal microbiota formed two different clusters based on the stage of SUS (early or advanced) in the juveniles. Skin microbiota was closely clustered despite the differences in the disease stage of the juveniles. Nevertheless, paired-PER-MANOVA showed significant differences between the intestinal samples of early and advanced-stages of SUS in cultured juveniles ($F = 5.44$, $R^2 = 0.28$, p = 0.001) and between the skin samples of early and advanced-stages SUS juveniles ($F = 5.21$, $R^2 = 0.29$, p = 0.025).

### Bacterial community composition in the intestine and skin samples of early-stage SUS juveniles

The intestinal bacterial composition of juveniles with early SUS stage, at class level, included *Alphaproteobacteria* (34.2±23.1%), *Bacteroidia* (25.7±18.8%), *Gammaproteobacteria* (16.0 ±14.8%) and *Campylobacteria* (12.3±10.4%) (Fig 4A). At lower taxonomical levels, the most abundant genera within *Alphaproteobacteria* were Unassigned_*Rhodobacteraceae* (22.3 ±23.7%), SM2D12 (5.9±12.5%), and *Shimia* (3.7±4.8%) (Fig 4B). Within *Bacteroidia*, high relative abundance was observed for Uncultured_*Cryomorphaceae* (18.1±14.6%) and *Flammeovirga* (2.4±2.7%). Additionally, *Vibrio* (10.2±10.4%) dominated within *Gammaproteobacteria*,

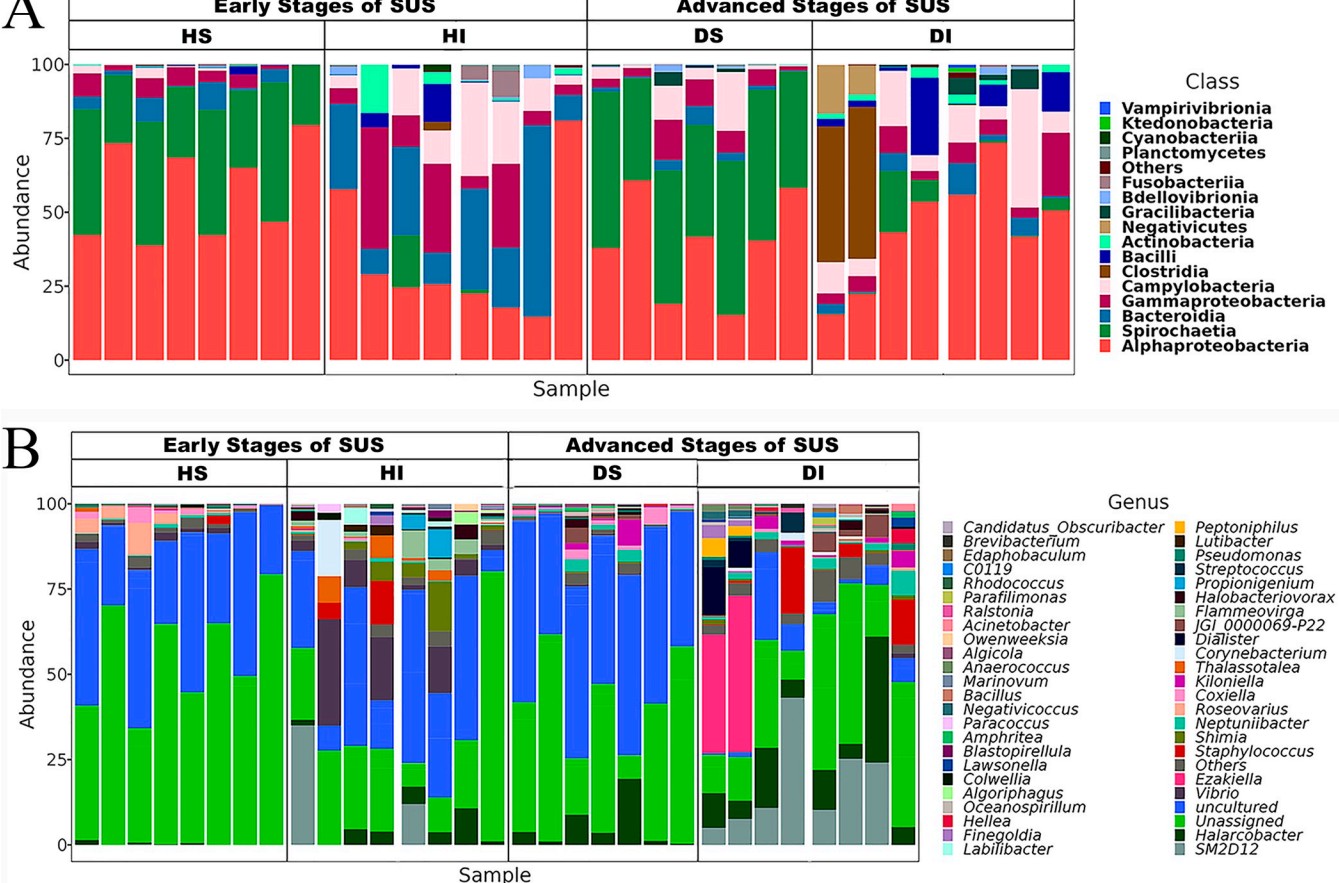

**Fig 4. Bacterial taxa abundance in intestine and skin samples of *I. badionotus* juveniles with SUS.** A) Classes of intestinal and skin bacterial ASVs. B) Genera of intestinal and skin bacterial ASVs. HS: early stage of SUS skin, HI: early stage of SUS intestine, DS: diseased skin, DI: diseased intestine. Most of the ASVs from intestine and skin samples could not be classified at the genus level. The genera with less than 1.0% of relative abundance were assigned to Others.

while *Halarcobacter* (3.8±3.3%) and *Thalassotalea* (2.5±3.1%) from *Campylobacteria* also exhibited high relative abundances. Interestingly, *Cyanobacteria* were only identified in the early stage of SUS samples.

Bacterial communities in the skin of early-stage SUS juveniles were primarily composed of *Alphaproteobacteria* (57.1±16.2%), *Spirochaetia* (33.4±11.0%) and *Gammaproteobacteria* (4.1±2.8%) (Fig 4A). The most abundant genera within *Alphaproteobacteria* were unassigned_*Rhodobacteraceae* (53.9±17.8%) and *Roseovarius* (2.7±3.2%) (Fig 4B). *Spirochaetia* included uncultured_*Spirochaetaceae* (33.4±11.0%) as the only identified genus. *Gammaproteobacteria* were represented by *Coxiella* (1.3±1.5%) and *Vibrio* (1.1±0.9%) as the genera with the highest relative abundance.

## Bacterial community composition in the intestine and skin samples of advanced-stage SUS juveniles

The intestinal microbiota composition of juveniles with advanced SUS stage showed variation compared to those with SUS in the early stage in their principal classes and relative abundance (Fig 4A). *Alphaproteobacteria* (44.6±18.6%) was the main class and presented higher relative abundance than juveniles with early SUS. An increase in the abundance of classes *Campylobacteria* (13.1±11.9%), *Clostridia* (12.2±22.6%), *Bacilli* (6.7±9.1%) and *Spirochaetia* (4.2±7.2%) was also observed in these diseased juveniles. *Gammaproteobacteria* (7.3±6.1%) and *Bacteroidia* (3.6±3.7%) showed significant decreases in their relative abundance. In addition, *Negativicutes* (3.3±6.4%) and *Gracilibacteria* (1.9±2.7%) were found exclusively in the intestine samples of juveniles with advanced SUS. At more specific levels, there was an increase in the relative abundance of SM2D12 (15.8±14.1%) and unassigned_*Rickettsiales* (7.6±14.1%), as well as a decrease in unassigned_*Rhodobacteraceae* (11.3±6.0%) within the *Alphaproteobacteria* class (Fig 4B). Furthermore, *Halarcobacter* (12.1±11.0%) also showed an increase in relative abundance. Finally, *Ezakiella* (10.0±18.8%), belonging to the *Clostridia* Class was notably abundant and found only in the intestine samples of juveniles with advanced SUS.

Regarding the skin samples of juveniles with advanced SUS, the main change at the class level compared to juveniles with early SUS was an increase in the relative abundance of *Spirochaetia* (44.7±7.5%), becoming the dominant class, followed by a decrease in *Alphaproteobacteria* (39.1±17.4%) and an increase in the relative abundance of *Gammaproteobacteria* (6.2±4.3%) and *Campylobacteria* (6.0±7.1%) (Fig 4A). At the genus level, uncultured_*Spirochaetaceae* had a higher relative abundance in the skin of juveniles with advanced SUS (44.7±7.5%) compared to juveniles with early SUS (Fig 4B). Among the *Alphaproteobacteria*, unassigned_*Rhodobacteraceae* (30.3±19.6%) and *Roseovarius* (0.07±0.08%) were less abundant compared to juveniles with early SUS. Meanwhile, unassigned_*Kiloniellaceae* (6.3±6.6%) and *Kiloniella* (1.5±2.8%) were the most abundant in juveniles with advanced SUS. Furthermore, *Coxiella* (1.9±1.5%) and *Neptuniibacter* (1.4±1.5%) were the most abundant genera within *Gammaproteobacteria*. Finally, *Halarcobacter* (5.4±6.8%) was the most abundant genus in the *Campylobacteria* class.

## Bacterial species associated with SUS in sea cucumbers

We conducted a species-level analysis of ASVs to recognize bacterial species associated with SUS in sea cucumbers, (S1 Table). The result shows that only 32.7% and 35.1% of the ASVs from intestinal samples to the early and advanced stages of SUS were taxonomically assigned to this level. On the other hand, the assigned percentages for skin samples were 2.7% and 3.3% for both early and advanced stages of SUS. Within the ASVs successfully assigned at the species level, we identified the presence of four bacterial species that have been previously reported as

potential SUS pathogens in sea cucumbers. These bacterial species were *Vibrio* sp., *Vibrio harveyi*, *Vibrio fortis*, and *Pseudoalteromonas spongiae* (S1 Table).

In the intestinal samples of cultured *I. badionotus* juveniles, we found *Vibrio* sp. in both SUS stages at a relative abundance of 0.95±0.7% in the early stage and 0.002±0.01% in the advanced stage. *V. harveyi* (0.10±0.2%) and *V. fortis* (1.0±1.8%) were exclusively present in samples from early-stage SUS juveniles. Moreover, *P. spongiae* was only identified in advanced SUS with a relative abundance of 0.01±0.03%.

Regarding skin samples, we found *Vibrio* sp. and *V. fortis* in both early SUS samples at a relative abundance of 0.30±0.3% and 0.08±0.2%, respectively. In the advanced SUS samples, their relative abundance was 0.3±0.04% and 0.04±0.05%, respectively. *V. harveyi* was only present in early SUS samples with a relative abundance of 0.03±0.09%. Finally, *P. spongiae* was exclusively found in samples of advanced SUS with a relative abundance of 0.03±0.05%.

## Differential abundance analysis between early and advanced SUS in *I. badionotus* juveniles

LEFSE analyses were performed to determine the bacterial genera with significant differences in abundance between both SUS stages in *I. badionotus* juveniles (Fig 5A and 5B). The analysis showed 23 and 21 bacterial genera in the intestinal and skin samples.

In the intestine of juveniles early SUS, uncultured_bacteria, *Vibrio*, *Thalassotalea*, and *Fammeovirga* were genera that contributed the most to significant differences in abundance. In contrast, *Halarcobacter*, followed by *Neptuniibacter*, represented the most abundant genera in juveniles with advanced SUS (Fig 5A). In addition, *Roseovarius* and *Vibrio* were the most abundant genera on the skin of juveniles with early SUS. In contrast, *Halacobacter* and *Killionella* were the most abundant genera in juveniles with advanced SUS (Fig 5B).

The LEFSE analysis identified that *Vibrio* and *Thalassotalea* exhibited significantly higher abundances in both the intestine and skin samples of juveniles with early SUS. In contrast, *Halarcobacter*, JGI_0000069_P22, *Kiloniella*, *Hellea*, *Oceanospirillum*, *Amphritea*, *Algicola*, *Oleibacter*, *Flavobacterium*, *Alteromonas*, and *Agarivorans* were found in both the intestine and skin samples of juveniles with advanced SUS.

## Discussion

In our attempt to determine the causative agent of the mortality of cultured juvenile *I. badionotus* in 2018, we discovered that some juvenile sea cucumbers that were initially considered healthy resulted in pathologies in their intestine and skin that were similar to those found in juveniles with an advanced SUS disease, but less severe. We also observed differences in the bacterial composition between juvenile sea cucumbers with early or advanced SUS. In the following paragraphs, we will provide detailed interpretations and explanations for these findings. Before proceeding, it is important to note that we had the opportunity to work with the very last survivors of this cultured *I. badionotus* population. Although the number of individual juveniles was low, the detailed histological and microbiological evidence presented here is relevant to increase our understanding of the etiological agents causing this disease and associated pathologies. This is especially important since this disease has been reported in other cultured sea cucumber populations worldwide [13, 19–21, 35–37].

Histological damage observed in juvenile *I. badionotus* suggests an inflammatory process. The damage includes the loss of epithelium in the intestinal mucosa and high levels of coelomocyte infiltration in both stages of SUS. The destruction or alteration of the epithelial cells in the gastrointestinal mucosa can facilitate the entry of microbial pathogens [38]. This can lead to an immune response that can produce an inflammatory process in the digestive tract [39].

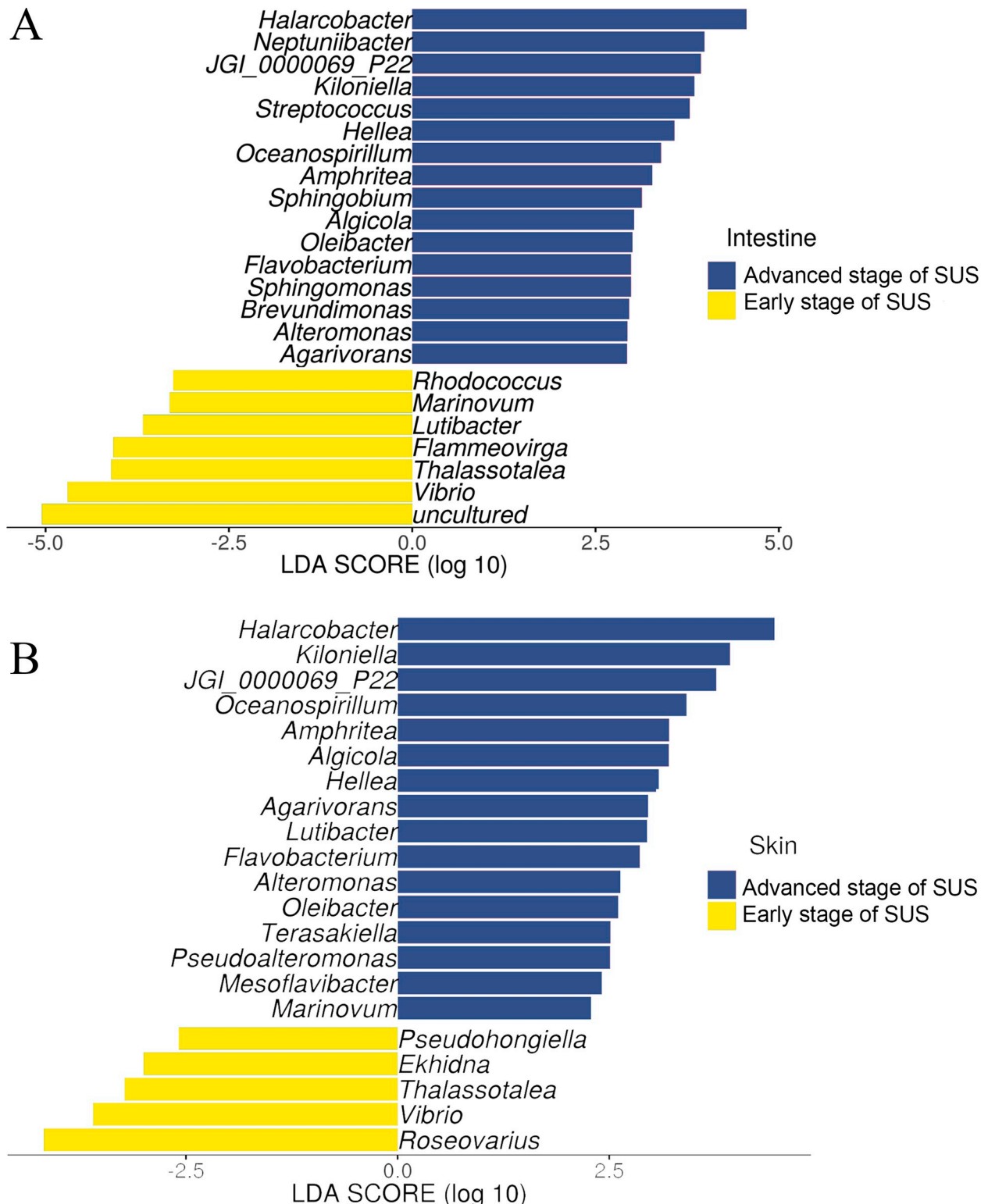

**Fig 5. LEFSE analysis of samples of both stages of SUS in cultured *I. badionotus* juveniles.** Linear discriminant analysis (LDA) scores (log 10) derived from LEFSE analysis of intestinal bacterial communities (A) and skin communities (B). A significance alpha of 0.05 was used for all biomarkers evaluated.

Additionally, coelomocytes play a significant role in the immune system of echinoderms by recognizing non-self-materials, fluid circulation, cytotoxic defense, clotting, encapsulation, and inflammatory responses. An increase in the abundance of coelomocytes has been observed in response to physical injury, stressors, and disease in sea cucumbers and other echinoderms [40–42]. Our results are similar to those reported in *A. japonicus* with SUS induced by *V. splendidus* [21]. Nevertheless, the bacteria or parasite clusters in the intestinal epithelium have not been reported previously [21].

The lesions observed in the first layers of the body wall, such as the loss of cuticle and epidermis, are characteristic signs of juveniles with advanced stages of SUS. In some cases, the lesions could be caused by opportunistic bacteria that take advantage of initial damage to develop into ulcers. The delicate nature of the cuticle and epidermis of sea cucumbers makes them susceptible to damage, whether caused by mechanical or physiological disorders [13, 43]. The liquefactive necrosis observed in the dermis of ulcerated juveniles is a pathology that has not been previously reported in histological sections of sea cucumbers with SUS. This necrosis is characterized by the partial or complete dissolution of dead tissue caused by microbial agents such as bacteria, fungi, viruses, and parasites [44].

Concerning microbial communities, our results showed no significant difference in alpha diversity between early and advanced stages of SUS in juveniles' intestines and skin microbiota. However, there was a significant difference in beta diversity between the two stages of SUS in both the intestine and skin samples of juveniles. Previous studies have identified decreased alpha diversity and changes in beta diversity in the gut microbiota of *A. japonicus* with SUS [19]. Our results from PCoA demonstrate that the composition of bacterial communities in the intestines and skin of *I. badionotus* juveniles with advanced SUS differed significantly from those with early SUS. This suggests that the stage of disease development plays a role in the changes observed in the composition of microbial communities, as seen in the intestinal microbiota of SUS-infected *A. japonicus* [21]. Furthermore, we observed that the composition of bacterial communities in the intestines differed more significantly than in the skin samples. These results may be related to the fact that similar intestinal histological damage was observed in both juveniles with early SUS, whereas in the skin samples, only one of the two organisms exhibited lesions similar to those observed in juveniles with advanced SUS.

The intestinal microbiota composition of early SUS *I. badionotus* juveniles was mainly composed of *Proteobacteria* (50.2%), *Bacteroidetes* (25.7%), *Campylobacterota* (12.3%) and *Actinobacteria* (3.1%), with *Firmicutes* present at a lower relative abundance of 2.8%. These results were similar at the Phylum level to those found in sea cucumbers of the same species kept in captivity (organisms collected from wild populations and kept in fiberglass tanks for six months) [12]. These authors reported that *Proteobacteria* (60.9%), *Actinobacteria* (11.6%), and *Bacteroidetes* (6.8%) were the predominant bacterial Phyla in the intestine. Nevertheless, the relative abundance of *Firmicutes* (12.9%) was higher than our results.

At the genus level, high abundances of *Vibrio* (10.2%), as well as the presence of *Shimia* (3.7%), *Halarcobacter* (Syn. *Arcobacter*) (3.8%), and *Thalassotalea* (2.5%), were observed in the intestine of early SUS juveniles. These results were similar to those observed in the hindgut of captive *I. badionotus*, where *Vibrio* (17.4%) and *Shimia* (11.7%) were the most abundant genera after *Ruegeria*. The low relative abundances of *Arcobacter* and *Thalassotalea* in the present study may be similar to those found in the captive sea cucumbers. This similarity may be attributed to the observation that the composition of bacterial species in the early stages of the initial hours of SUS disease development is the same as that of healthy organisms [21, 36]. This similarity also may be related to the time that sea cucumbers have spent in captivity (six months), which can influence the composition of the intestinal microbiota. The gut microbiota of sea cucumbers is influenced by both intrinsic factors such as genetics, immune status, and

life stage, and extrinsic factors such as diet and environmental conditions [45–47]. Significant changes in the gut microbiota have been observed in *Holothuria glaberrima* between wild sea cucumbers and those maintained in aquaria for just three days [45]. *Vibrio* is a genus that has been found in high abundance in the intestines of healthy *A. japonicus* and *H. glaberrima* [45, 48–50]. Therefore, this genus appears to be part of the normal microbiota of sea cucumbers.

We observed differences in the structure of the bacterial community in the intestines of cultured *I. badionotus* juveniles between the early and advanced stages of SUS. In the early stage of SUS, the Phyla *Proteobacteria* (Alpha and *Gammaproteobacteria*), *Bacteroidota* (synonym of *Bacteroidetes*) and *Campylobacterota* (*Campylobacteria*) were numerically dominant. In contrast, in the intestinal samples of advanced-stage SUS juveniles, classes of the Phyla *Proteobacteria* and *Firmicutes* (*Clostridia*, *Bacilli*, and *Negativicutes*) were more abundant. These findings are consistent with recent evidence that changes in the intestinal microbiota are associated with the development of SUS in sea cucumbers [19–21].

Our study found that *Proteobacteria* and *Firmicutes* were more abundant in the intestines of juveniles with advanced SUS, while the abundance of *Bacteroidetes* drastically decreased. Similarly, a higher abundance of *Proteobacteria* and *Firmicutes* groups has been reported in the intestine of SUS-diseased sea cucumbers compared to healthy ones [19, 21]. It has also been noted that during the development of the disease, *Proteobacteria* decreased in abundance in the first 48 h and increased after 96 h, while *Firmicutes* exhibited the opposite pattern, increasing in abundance at 48 h with a subsequent drastic decrease at 96 h [21].

Bacterial communities in the skin of juveniles with both early and advanced stages of SUS were found to be numerically dominated by *Proteobacteria* and *Spirochaeta*. However, there were differences in the relative abundance at the class level. In juveniles with early SUS, *Alphaproteobacteria* were the most dominant class, followed by *Spirochaetia* and *Gammaproteobacteria*. In contrast, in juveniles with advanced SUS, the relative abundance of these last two classes increased while that of *Alphaproteobacteria* decreased. Similar to our findings, previous research has also reported a dominance of *Proteobacteria* in both healthy and diseased *Holothuria scabra* skin samples, with a higher proportion of these bacteria in ulcerated organisms [13].

In our study, we identified various bacterial species associated with SUS. These species include *Vibrio* sp., *V. harveyi*, *V. fortis*, and *P. spongiae*. However, we found that they were present in very low abundances of less than 1% in both intestinal and skin samples from juveniles in both disease stages. Therefore, these species were not considered causative agents of SUS in our study. Although these species have been associated as potential pathogens of SUS in sea cucumbers such *Holothuria arguinensis*, *H. scabra*, and *A. japonicus*, when were isolated from ulcers of diseased organisms, only one study has successfully induced SUS by injecting *V. harveyi* into healthy sea cucumbers [13, 15, 16]. However, this study also identified other species capable of inducing the disease [48]. Therefore, there is no evidence that *V. harveyi* is the sole cause of SUS. Recent studies have concluded that the disease appears to be multifactorial, caused by biotic or abiotic factors or by a combination of both [13].

The LEFSE analysis revealed that the genera *Vibrio* and *Thalassotalea* showed a significant association with juveniles with early SUS. These findings were similar to those found in the microbiota of healthy organisms. At this stage of the disease, the microbiota is usually similar to that of healthy organisms. However, the bacterial composition at the genus level (seen in Fig 4B) showed a decrease in the abundance of these bacterial genera in juveniles with advanced SUS, particularly in the intestine samples. Our results differ from those previously reported in the literature, where bacteria from the *Vibrio* genus (*Vibrio* spp., *V. alginolyticus*, and *Vibrio splendidus*) were identified as the primary causes of inducing SUS in sea cucumber [21, 48–51]. Different species of *Vibrio* can be of benefit to animals, and not just act as pathogens.

There are examples of using *Vibrio* strains as probiotics in aquaculture [52–54]. Therefore, our results suggest that a decrease in the abundance of *Vibrio* during SUS-disease progression could be related to a reduction of beneficial *Vibrio* strains in the intestine of *I. badionotus* juveniles due to stress conditions in the host.

On the other hand, *Thalassotalea* is a predominant bacterial found in the intestinal microbiota of *A. japonicus* [58]. However, it has also been isolated from various marine environments such as sediments, seawater, corals, oysters, and even in marine recirculating aquaculture systems [55]. Therefore, its role may be related to the recycling of organic matter and nutrients in the culture environment, as previous studies have suggested [55, 56]. Nevertheless, the progression of SUS appears to have a negative impact on the abundance of species within this genus.

In Fig 4B, we can observe a significant increase in the abundance of *Halarcobacter* (Syn. *Arcobacter*) from the early to the advanced stage of SUS in cultured juveniles of *I. badionotus*. It's worth noting that some species of *Arcobacter* are pathogenic and considered emerging enteropathogens in both humans and livestock [57]. In holothuroids, *Arcobacter* has been isolated from the intestine and the skin ulcers of *A. japonicus* and *H. scabra* [15, 58, 59].

Currently, no scientific evidence indicating a decrease in the *Vibrio* bacterial genus during SUS. *Vibrio* is a bacterium typically found in sea cucumber species such as *I. badionotus, A. japonicus,* and *H. glaberrima*, and is known to be beneficial for them. However, a decrease in the abundance of *Vibrio* can have significant consequences because it allows for the colonization and proliferation of potentially pathogenic and opportunistic bacteria, such as *Halarcobacter* (syn. *Arcobacter*), that has also been identified as part of the microbiota of healthy sea cucumber organisms [12, 58].

A similar phenomenon has been observed in the gut microbiota of juvenile *A. japonicus*, where the antibiotics significantly decreased the abundance of beneficial bacteria such as *Vibrio* and *Thalassotalea*, which were dominant bacterial genera in their microbiota along with *Halarcobacter* (syn. *Arcobacter*). However, the decrease in the abundance of these bacterial genera led to a significant increase in the abundance of *Halarcobacter* (syn. *Arcobacter*, which increases the risk of host infection [58]. Furthermore, an increase in the abundance of *Arcobacter* spp. has been reported in both internal and exposed tissues of diseased or stressed fish, oysters, and shrimp [60], suggesting this bacterium´s pathogenic potential due to microbiota alterations.

In addition to *Halarcobacter* (syn. *Arcobacter*), other pathogenic bacteria belonging to the genera *Hellea, Algicola, Flavobacterium* and *Agarivorans* were identified in both tissues of *I. badionotus* juveniles (Fig 5). Studies have shown that a significantly higher relative abundance of *Hellea* is present in diseased algae such as *Saccharina japonicus* and *Delisea pulchra* compared to the healthy ones. Therefore, it is believed that *Hellea* plays an important role in the progression of bleaching disease in farmed *S. japonica* [61]. On the other hand, *Algicola* is another genus of bacteria that has been isolated from lesions on the seaweed *Laminaria japonica*, but its presence in lesions is more related to a bacteriolytic effect [62]. Within the genus *Flavobacterium*, *F. psychrophilum* is a known opportunistic pathogenic species responsible for Bacterial Coldwater Disease (BCWD) and Rainbow Trout Fry Syndrome (RTFS) [60]. Finally, *Agarivorans albus* has been identified as one of the pathogens that produce bleached disease in the marine macroalga *Gracilaria lemaneiformis* [63].

Given that *Halarcobacter* (syn. *Arcobacter*) was not the only pathogenic bacterial genus identified in our samples, we cannot be certain that species within this genus are the only ones responsible for the SUS outbreak in the *I. badionotus* culture. However, we observed that *Halarcobacter* exhibited the greatest increase in abundance from the early to the advanced SUS stage. Therefore, we recommend that special attention should be given to species within this genus in *I. badionotus* aquaculture.

## Conclusion

The *I. badionotus* juveniles, which were initially considered healthy due to the absence of skin lesions, resulted in an early stage of SUS since similar pathologies with lower damage were observed in the intestinal and skin samples compared to the advanced stage. These histological changes and the differences observed in the composition and structure of the microbial communities between the two stages of SUS suggest that the disease was progressing. Our results suggested that a decrease in the abundance of *Vibrio* and an increase in *Halarcobacter* (syn. *Arcobacter*) could be related to the development of SUS disease in *I. badionotus* juveniles. Therefore, it is necessary to implement longitudinal monitoring of microbial communities in culture systems, including those in the seawater used for culture, to implement timely interventions, such as adjustment of culture conditions or administration of probiotics. This is especially important in the case of bacterial genera reduced during disease progression, which in turn could prevent the spread of SUS in the *I. badionotus* culture.

## Supporting information

**S1 Table. Relative abundance of taxonomic assignment of ASVs from all samples.**
(XLSX)

**S2 Table. Species richness and alpha diversity indices from all samples.**
(XLSX)

## Acknowledgments

The authors would like to thank Gregory Arjona for the help with the histological processing and Linda Marmolejo for figures editing. We also thank. Pedro Tec Tec, and Gilmer Uicab Uc for their support in producing and maintaining experimental organisms.

## Author Contributions

**Conceptualization:** Karen A. Arjona-Cambranes, Miguel A. Olvera-Novoa, Víctor M. Vidal-Martínez, José Q. García-Maldonado.

**Data curation:** Daniel Cerqueda-García.

**Formal analysis:** Karen A. Arjona-Cambranes, Daniel Cerqueda-García.

**Funding acquisition:** Víctor M. Vidal-Martínez, José Q. García-Maldonado.

**Investigation:** Karen A. Arjona-Cambranes.

**Methodology:** Karen A. Arjona-Cambranes.

**Project administration:** Víctor M. Vidal-Martínez, José Q. García-Maldonado.

**Resources:** Miguel A. Olvera-Novoa, Víctor M. Vidal-Martínez, José Q. García-Maldonado.

**Software:** Daniel Cerqueda-García.

**Supervision:** Víctor M. Vidal-Martínez, José Q. García-Maldonado.

**Validation:** Karen A. Arjona-Cambranes, Madeleine G. Arjona-Torres.

**Visualization:** Karen A. Arjona-Cambranes.

**Writing – original draft:** Karen A. Arjona-Cambranes.

**Writing – review & editing:** Miguel A. Olvera-Novoa, Daniel Cerqueda-García, Madeleine G. Arjona-Torres, M. Leopoldina Aguirre-Macedo, Víctor M. Vidal-Martínez, José Q. García-Maldonado.

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
