## [Decision Letter · Decision Letter 0]

26 Dec 2023

PONE-D-23-33549Characterization of microbiota and histology of cultured sea cucumber Isostichopus badionotus juveniles during an outbreak of skin ulceration syndrome.PLOS ONE

Dear Dr. García-Maldonado,

Thank you for submitting your manuscript to PLOS ONE. After careful consideration, we feel that it has merit but does not fully meet PLOS ONE’s publication criteria as it currently stands. Therefore, we invite you to submit a revised version of the manuscript that addresses the points raised during the review process.

We look forward to receiving your revised manuscript.

Kind regards,

Tomoo Sawabe

Academic Editor

PLOS ONE

Journal Requirements:

"This research was supported by the Mexican National Council for Humanities, sciences and Technologies- CONAHCYT – Mexican Ministry of Energy- Hydrocarbon Fund, project 201441; Fomix-CONAHCYT project No.246841. CONAHCYT awarded KAAC with doctoral scholarship No. 775910. This is a contribution of the Gulf of Mexico Research Consortium (CIGoM). "

"This research was supported by the Mexican National Council for Humanities, sciences and Technologies- CONAHCYT – Mexican Ministry of Energy- Hydrocarbon Fund, project 201441; Fomix-CONAHCYT project No.246841. CONAHCYT awarded KAAC with doctoral scholarship No. 775910. This is a contribution of the Gulf of Mexico Research Consortium (CIGoM).

The authors would like to thank HT. Gregory Arjona for the help with the histological processing, and MSc. Linda Marmolejo for figures editing. We also thank Biol, Pedro Tec Tec and Biol. Gilmer Uicab Uc for their support in producing and maintaining experimental organisms. "

"This research was supported by the Mexican National Council for Humanities, sciences and Technologies- CONAHCYT – Mexican Ministry of Energy- Hydrocarbon Fund, project 201441; Fomix-CONAHCYT project No.246841. CONAHCYT awarded KAAC with doctoral scholarship No. 775910. This is a contribution of the Gulf of Mexico Research Consortium (CIGoM). "

5. Please note that in order to use the direct billing option the corresponding author must be affiliated with the chosen institute. Please either amend your manuscript to change the affiliation or corresponding author, or email us at plosone@plos.org with a request to remove this option.

6. We note that your Data Availability Statement is currently as follows: [All relevant data are within the manuscript and its Supporting Information files.]

7. Please amend either the abstract on the online submission form (via Edit Submission) or the abstract in the manuscript so that they are identical.

Additional Editor Comments:

Associate Editor

1. Authors are requested to perform ASV level analyses to mine SUS associated bacterial species.

2. Authors are requested to describe association of eukaryotic microbes to SUS.

3. Authors are requested to use Italic for all taxonomic names even in higher taxa.

Reviewer 1

While the study compares microbiotas in sea cucumbers affected by Sea Cucumber Skin Ulceration Syndrome (SUS), it would benefit from a control group and a larger sample size to strengthen its analysis. Additionally, the novelty of this study should be clarified, as SUS-related microbial changes have been extensively studied before.

Line 88, can you provide the details of filtered seawater? (Nature or artificial, sterilized or not)

Line 90, I didn't quite understand, the water was renewed at 60% of new water?

Line 108, sample size might be too small, I think it should be at least 3 samples per group.

Line 109, no control/ healthy group?

Line 119, I am wondering the stage of SUS of these four juveniles, you mentioned about two of them were healthy when you collecting the samples, how about the others? If they were in disease, how could you collected the diseased microbiota after washing the surfaces?

Line 125, I didn't understand the " three from the second one", do you mean the stage between early to advanced?

Line 260, since you compared microbiotas between different sections and individuals, how do you know the differences were caused by disease or individual variation?

Line 391, it is not a complete sentence.

Line 419-426, since you mentioned about the differences of microbiome caused by culture condition and section of intestine, how could you know the change of specific bacteria is associated with the SUS?

Line 454, can you explain why Vibrio decreased in advanced stage of SUS in your study?

Line 499, could you add some discussion about how can you prevent the spread of the disease according to your result?

Reviewers' comments:

Reviewer's Responses to Questions

**Comments to the Author**

1. Is the manuscript technically sound, and do the data support the conclusions?

Reviewer #1: No

2. Has the statistical analysis been performed appropriately and rigorously? 

Reviewer #1: Yes

3. Have the authors made all data underlying the findings in their manuscript fully available?

Reviewer #1: Yes

4. Is the manuscript presented in an intelligible fashion and written in standard English?

Reviewer #1: Yes

5. Review Comments to the Author

Reviewer #1: While the study compares microbiotas in sea cucumbers affected by Sea Cucumber Skin Ulceration Syndrome (SUS), it would benefit from a control group and a larger sample size to strengthen its analysis. Additionally, the novelty of this study should be clarified, as SUS-related microbial changes have been extensively studied before.

Line 88, can you provide the details of filtered seawater? (Nature or artificial, sterilized or not)

Line 90, I didn't quite understand, the water was renewed at 60% of new water?

Line 108, sample size might be too small, I think it should be at least 3 samples per group.

Line 109, no control/ healthy group?

Line 119, I am wondering the stage of SUS of these four juveniles, you mentioned about two of them were healthy when you collecting the samples, how about the others? If they were in disease, how could you collected the diseased microbiota after washing the surfaces?

Line 125, I didn't understand the " three from the second one", do you mean the stage between early to advanced?

Line 260, since you compared microbiotas between different sections and individuals, how do you know the differences were caused by disease or individual variation?

Line 391, it is not a complete sentence.

Line 419-426, since you mentioned about the differences of microbiome caused by culture condition and section of intestine, how could you know the change of specific bacteria is associated with the SUS?

Line 454, can you explain why Vibrio decreased in advanced stage of SUS in your study?

Line 499, could you add some discussion about how can you prevent the spread of the disease according to your result?

6. PLOS authors have the option to publish the peer review history of their article (what does this mean?). If published, this will include your full peer review and any attached files.

Reviewer #1: **Yes: **Juanwen YU

---

## [Author Response · Author response to Decision Letter 0]

8 Feb 2024

Associate Editor

1. Authors are requested to perform ASV level analyses to mine SUS associated bacterial species.

We recognize the importance of identifying bacterial species associated with SUS. Therefore, we have incorporated an analysis of ASVs at the species level into the paper, complemented by a supplementary table. This section has been entitled “Bacterial species associated with SUS in sea cucumbers” (see lines 344-363).

 Moreover, we included a paragraph about this information in the discussion section (see lines 477-487). 

2. Authors are requested to describe association of eukaryotic microbes to SUS.

We recognize the importance of characterizing microeukaryotic communities associated with the SUS. However, we have encountered limitations in identifying these communities during an additional analysis that we previously conducted. In that exercise, we analyzed the eukaryotic communities in auricularia larvae of Isostichopus badionotus based on 18S rRNA sequencing, however we found that more than 94% of the sequences were taxonomically assigned to the sea cucumber. Consequently, we recognize the need for the development and implementation of blocking primers that inhibit the amplification of I. badionotus DNA to achieve a broader representation of these communities in the samples. As a result of these considerations, the present work focused exclusively on the characterization of bacterial communities.

3. Authors are requested to use Italic for all taxonomic names even in higher taxa.

Changes were made to all taxonomic groups as requested.

Reviewer 1

1. While the study compares microbiotas in sea cucumbers affected by Sea Cucumber Skin Ulceration Syndrome (SUS), it would benefit from a control group and a larger sample size to strengthen its analysis. Additionally, the novelty of this study should be clarified, as SUS-related microbial changes have been extensively studied before.

We recognize the importance of having a control group and a larger sample size. Initially, we selected two juveniles as control group because they appeared to be healthy. However, upon histological analysis, these juveniles, despite showing no visible skin lesions, exhibited tissue damage similar to that observed in the histological samples of the ulcerated juveniles (see lines 112 to 114). Consequently, we inferred that they might be in the early stage of SUS disease, which could explain the absence of visible lesions at that point. We did not have access to more juvenile sea cucumbers, due to high mortality of these organisms. By this reason, it was not possible to include a control group.

Regarding sample size, the samples used in this study came from the latest group of juveniles obtained under culture conditions in our facilities. At that point, the culture of I. badionotus had very limited number of specimens, and its population was significantly reduced due to the occurrence of SUS disease, which initially constrained our sample size. To date, the progress of our experimental culture has been negatively affected by health issues of the early life stages of sea cucumbers. These challenges have prevented us from acquiring new juveniles, thus limiting our ability to expand the sample size or acquire new specimens that could be used as a control group. We highlight the importance of these samples in lines 105-106.

While we acknowledge the extensive study of SUS disease, the novelty of our research lies in introducing, for the first time, the alterations that occur in the bacterial communities associated with SUS in the cultured sea cucumber species I. badionotus. 

2. Line 88, can you provide the details of filtered seawater? (Nature or artificial, sterilized or not)

This information was included in lines 88-92. “filled with natural seawater obtained from a 24-m deep beach well. The seawater was filtered using a sand filter and treated with ultraviolet (UV) radiation. Furthermore, the sand substrate at the bottom of the containers was also subjected to UV radiation treatment. The tanks were part of a closed recirculating system formed by a sand filter, a biological filter, continuous UV radiation (2 x 40 W), and a water chiller fixed at 25 °C.”

3. Line 90, I didn't quite understand, the water was renewed at 60% of new water?

This information was clarified in line 93-94. “To remove faces and unconsumed food, 60% of the seawater was replaced every two days”.

4. Line 108, sample size might be too small, I think it should be at least 3 samples per group.

We recognize that sample size was small, but it was due to the reasons mentioned in question 1 above. Nevertheless, additional sample replications were conducted to address the limited sample size and enhance the robustness of the study (see lines 125-130).

5. Line 109, no control/ healthy group?

This information was addressed in the first question above.

6. Line 119, I am wondering the stage of SUS of these four juveniles, you mentioned about two of them were healthy when you collecting the samples, how about the others? If they were in disease, how could you collected the diseased microbiota after washing the surfaces?

The information about the remaining two juveniles was clarified in line 114-116. “The other two juveniles exhibited signs of disease, manifested by visible ulcers on their skin, and were classified as being in the advanced stage of the disease”.

The surface washing was conducted to eliminate bacterial communities not specific to the skin sample, which could introduce interference as potential contamination. This contamination might arise from water and organic matter accumulated in the culture tanks (refer to line 123). This methodology has been previously presented and discussed by Tangestani and Kunzmann, 2019 (reference 15). The ulcers observed on the skin of our juveniles extended down to the dermis, making it unlikely that the communities present at that level were affected by the superficial washings. Moreover, these washes were essential to prevent contamination of internal organs during the dissection of the organisms.

7. Line 125, I didn't understand the " three from the second one", do you mean the stage between early to advanced?

We rewrite this paragraph to make easier to understand the ideas. See lines 127-130, “For molecular analysis, we collected eight skin samples from juveniles in the early stage of SUS (four samples from each juvenile). Additionally, seven skin ulcer samples were collected from juveniles in the advanced stage of SUS (three ulcer samples from one organism and four from the other one)”.

8. Line 260, since you compared microbiotas between different sections and individuals, how do you know the differences were caused by disease or individual variation?

While we physically separated the anterior and posterior parts of the gut during sampling, we did not conduct a specific analysis of the microbiota between these gut sections. Instead, the analysis considered both sections as a whole. We clarified this information in the lines 176-178. This approach aimed to represent the overall bacterial diversity of the sea cucumber digestive tract to facilitate a comparison between the early and advanced stages of SUS disease). Unfortunately, as mentioned above, it was not possible to increase the sample size for this study.

9. Line 391, it is not a complete sentence.

Thank you for your comment, this information was not informative, and it was deleted. Additionally, we clarified the information in line 421-422. “This necrosis is characterized by dead tissue's partial or complete dissolution resulted from infection with microbial agents (bacteria, fungi, viruses, and parasites).

10. Line 419-426, since you mentioned about the differences of microbiome caused by culture condition and section of intestine, how could you know the change of specific bacteria is associated with the SUS?

The information was clarified in line 445-453. “This similarity may be attributed to the observation that the bacterial species composition in the early stages or initial hours of SUS disease development is the same as that of healthy organisms [21,36]. This similarity may also be related to the time that sea cucumbers have spent in captivity (six months), since the composition of the intestinal microbiota is influenced by both intrinsic factors (genetics, immune status, life stage) and extrinsic factors (diet and environmental conditions) [45-47]. The same pattern has been observed in Holothuria glaberrima, where significant changes in the intestinal microbiota were observed between wild cucumbers and those maintained in aquaria for only three days [45]”.

We cannot definitively establish a link between the specific bacterial changes observed in our study and the SUS disease, primarily due to the absence of a control group and the lack of microbiota analysis by intestinal sections. Nevertheless, it is apparent that the microbiota exhibited significant differences between the two presented juvenile categories. Moreover, the outcomes of histological analyses indicating variations in the severity of observed pathologies, provided compelling evidence to suggest that changes in the bacterial communities are highly likely associated with the progression of the disease. For this reason, we have decided to remove the text on line 453: “Moreover, differences can also be influenced by the specific section of the intestine, its anatomical composition and its function [12]”, to avoid confusion.

11. Line 454, can you explain why Vibrio decreased in advanced stage of SUS in your study?

A discussion of the possible reasons for the decrease in Vibrio abundance in our study has been added in the lines 494-496. “Different species of Vibrio can be of benefit for animals, and not only pathogens. Actually, in aquaculture there are examples of the use of Vibrio strains as probiotics [52–54]. Therefore, our results suggest that a decrease of the abundance of Vibrio during SUS-disease progression could be related to a reduction of beneficial Vibrio strains in the intestine of I. badionotus juveniles, due to stress conditions into the host”.

12. Line 499, could you add some discussion about how can you prevent the spread of the disease according to your result?

The discussion was added in lines 541-546. “Therefore, the implementation of longitudinal monitoring of microbial communities in culture systems, including those in the seawater used for culture, is necessary to implement timely interventions, such as adjustment of culture conditions or administration of probiotics. This is especially important in the case of bacterial genera reduced during disease progression, which in turn could prevent the spread of SUS in the I. badionotus culture.”

---

## [Decision Letter · Decision Letter 1]

12 Mar 2024

PONE-D-23-33549R1Characterization of microbiota and histology of cultured sea cucumber Isostichopus badionotus juveniles during an outbreak of skin ulceration syndrome.PLOS ONE

Dear Dr. García-Maldonado,

Thank you for submitting your manuscript to PLOS ONE. After careful consideration, we feel that it has merit but does not fully meet PLOS ONE’s publication criteria as it currently stands. Therefore, we invite you to submit a revised version of the manuscript that addresses the points raised during the review process.

We look forward to receiving your revised manuscript.

Kind regards,

Tomoo Sawabe

Academic Editor

PLOS ONE

Reviewers' comments:

Reviewer's Responses to Questions

**Comments to the Author**

1. If the authors have adequately addressed your comments raised in a previous round of review and you feel that this manuscript is now acceptable for publication, you may indicate that here to bypass the “Comments to the Author” section, enter your conflict of interest statement in the “Confidential to Editor” section, and submit your "Accept" recommendation.

Reviewer #1: All comments have been addressed

2. Is the manuscript technically sound, and do the data support the conclusions?

Reviewer #1: Partly

3. Has the statistical analysis been performed appropriately and rigorously? 

Reviewer #1: Yes

4. Have the authors made all data underlying the findings in their manuscript fully available?

Reviewer #1: Yes

5. Is the manuscript presented in an intelligible fashion and written in standard English?

Reviewer #1: Yes

6. Review Comments to the Author

Reviewer #1: 1.I understand the difficulty in collecting the control groups. However, I suggest adding a discussion about the microbiota of healthy I. badionotus. This addition would help clarify the differences in bacterial groups between diseased and healthy I. badionotus, and provide insight into which bacteria may be inducing the disease.

2. Regarding the ASV analysis, could you consider conducting a more in-depth analysis? For example, you could focus on significantly changed ASVs and the most abundant ASVs at specific stages.

3. I still have some confusion regarding the role of Vibrio in this context. Have any reference studies shown a decrease in Vibrio during the SUS? Clarifying this point would strengthen the discussion surrounding Vibrio's involvement in the disease process.

7. PLOS authors have the option to publish the peer review history of their article (what does this mean?). If published, this will include your full peer review and any attached files.

Reviewer #1: **Yes: **Juanwen Yu

---

## [Author Response · Author response to Decision Letter 1]

13 Apr 2024

Reviewer 1

1. I understand the difficulty in collecting the control groups. However, I suggest adding a discussion about the microbiota of healthy I. badionotus. This addition would help clarify the differences in bacterial groups between diseased and healthy I. badionotus and provide insight into which bacteria may be inducing the disease.

We recognize the importance of adding information about the microbiota of healthy I. badionotus. However, we consider that even with this additional information, the findings of our study do not allow us conclusively to determine which bacteria induce the disease. This is because, as mentioned in lines 58-60, this disease is multifactorial and can be caused by biotic and abiotic factors, or a combination of both. Infection assays, such as those performed by Delroisse et al. (2020), are needed to confirm whether the proposed bacteria cause the disease.

 Only two studies have reported the microbiota composition of healthy I. badionotus in the scientific literature. The first, by Luna-Fontalvo et al. (2014), focused on characterizing the diversity of culturable bacteria from the skin and gut of I. badionotus collected in Colombia. However, this study does not present the role of these organisms in sea cucumbers and does not provide information on the relationship between these bacteria and the disease.

The second study by Quintanilla-Mena et al. (2022) used 16s rRNA gene sequencing to characterize the microbiota in different sections of the digestive tract of I. badionotus maintained in captivity for six months in Telchac, Yucatan. This study shares methodological and geographic similarities with our research. However, their samples were collected from sea cucumbers maintained in captivity for a period of time, while ours are from fully cultured sea cucumbers. Due to these differences in the origin of the samples, differences in the microbiota composition are likely to be observed. Nevertheless, we decided to base our discussion of the microbiota of healthy I. badionotus on their results, since the study of Quintanilla-Mena et al. (2022) provides the closest comparison to the microbiota of a healthy I. badionotus. Furthermore, we extended the discussion to the genus level to better identify possible changes or similarities in gut microbiota composition (see lines 436-459).

“The intestinal microbiota composition of our early SUS I. badionotus juveniles was mainly composed of Proteobacteria (50.2%), Bacteroidetes (25.7%), Campylobacterota (12.3%) and Actinobacteria (3.1%), with Firmicutes present at a lower relative abundance of 2.8%. These results were similar at the Phylum level to those found in sea cucumbers of the same species kept in captivity (organisms collected from wild populations and kept in fiberglass tanks for six months) [12]. These authors reported that Proteobacteria (60.9%), Actinobacteria (11.6%), and Bacteroidetes (6.8%) were the predominant bacterial Phyla in the intestine. Nevertheless, the relative abundance of Firmicutes (12.9%) was higher than our results.

At the genus level, high abundances of Vibrio (10.2%), as well as the presence of Shimia (3.7%), Halarcobacter (syn. Arcobacter) (3.8%), and Thalassotalea (2.5%), were observed in the intestine of early SUS juveniles. These results were similar to those observed in the hindgut of captive I. badionotus, where Vibrio (17.4%) and Shimia (11.7%) were the most abundant after Ruegeria. The low relative abundances of Arcobacter and Thalassotalea in the present study may be similar to those found in the captive sea cucumbers. This similarity may be attributed to the observation that the composition of bacterial species in the early stages of the initial hours of SUS disease development is the same as that of healthy organisms [21,36]. This similarity may be attributed to the observation that the composition of bacterial species in the early stages of the initial hours of SUS disease development is the same as that of healthy organisms [21,36]. This similarity also may be related to the time that sea cucumbers have spent in captivity (six months), which can influence the composition of the intestinal microbiota. The gut microbiota of sea cucumbers is influenced by both intrinsic factors such as genetics, immune status, and life stage, and extrinsic factors such as diet and environmental conditions [45–47]. Significant changes in the gut microbiota have been observed in Holothuria glaberrima between wild sea cucumbers and those maintained in aquaria for just three days [45]. Vibrio is a genus that has been found in high abundance in the intestines of healthy A. japonicus and H. glaberrima [45,48–50]. Therefore, this genus appears to be part of the normal microbiota of sea cucumbers.”

2. Regarding the ASV analysis, could you consider conducting a more in-depth analysis? For example, you could focus on significantly changed ASVs and the most abundant ASVs at specific stages.

We agree that focusing on the ASVs that have undergone significant changes at each disease stage is important. In fact, the LEFSE analysis was performed to compare the bacterial genera producing changes in the composition and structure of the microbiota of cultured I. badionotus at both disease stages (early and late).

In this analysis, we observed that Vibrio and Thalassotalea are two relevant genera in the early stage of SUS, both in the skin and intestinal microbiota. However, the identification of Vibrio in the gut microbiota of I. badionotus supports the association of this genus with the early stage of the disease. 

On the other hand, Halarcobacter (syn. Arcobacter) represents the bacterial genus with the highest significant association in both the skin and intestine of juvenile I. badionotus during the advanced stage of SUS. However, as we indicated in lines 38-39, this is not the only bacterial genus that could be involved in the development of the disease, since 15 other genera were identified as contributing to the change in the composition and structure of the microbiota during this stage. Nevertheless, the increase in the abundance of Halarcobacter (syn. Arcobacter) from the early to the late stage of SUS suggests that they play an important role in developing the disease.

3. I still have some confusion regarding the role of Vibrio in this context. Have any reference studies shown a decrease in Vibrio during the SUS? Clarifying this point would strengthen the discussion surrounding Vibrio's involvement in the disease process.

To answer this question, we added the following text in lines 520-535. 

“Currently, no scientific evidence indicating a decrease in the Vibrio bacterial genus during SUS. Vibrio is a bacterium typically found in sea cucumber species such as I. badionotus, A. japonicus, and H. glaberrima, and is known to be beneficial for them. However, a decrease in the abundance of Vibrio can have significant consequences because it allows for the colonization and proliferation of potentially pathogenic and opportunistic bacteria, such as Halarcobacter (syn. Arcobacter), that has also been identified as part of the microbiota of healthy sea cucumber organisms [12,58].

A similar phenomenon has been observed in the gut microbiota of juvenile A. japonicus, where the antibiotics significantly decreased the abundance of beneficial bacteria such as Vibrio and Thalassotalea, which were dominant bacterial genera in their microbiota along with Halarcobacter (syn. Arcobacter). However, the decrease in the abundance of these bacterial genera led to a significant increase in the abundance of Halarcobacter (syn. Arcobacter), which increases the risk of host infection [58]. Furthermore, an increase in the abundance of Arcobacter spp. has been reported in both internal and exposed tissues of diseased or stressed fish, oysters, and shrimp [63], suggesting this bacterium´s pathogenic potential due to microbiota alterations.”

---

## [Decision Letter · Decision Letter 2]

25 Apr 2024

Characterization of microbiota and histology of cultured sea cucumber Isostichopus badionotus juveniles during an outbreak of skin ulceration syndrome.

PONE-D-23-33549R2

Dear Dr. García-Maldonado,

We’re pleased to inform you that your manuscript has been judged scientifically suitable for publication and will be formally accepted for publication once it meets all outstanding technical requirements.

Kind regards,

Tomoo Sawabe

Academic Editor

PLOS ONE

Additional Editor Comments (optional):

Reviewers' comments:

Reviewer's Responses to Questions

**Comments to the Author**

1. If the authors have adequately addressed your comments raised in a previous round of review and you feel that this manuscript is now acceptable for publication, you may indicate that here to bypass the “Comments to the Author” section, enter your conflict of interest statement in the “Confidential to Editor” section, and submit your "Accept" recommendation.

Reviewer #1: All comments have been addressed

2. Is the manuscript technically sound, and do the data support the conclusions?

Reviewer #1: Yes

3. Has the statistical analysis been performed appropriately and rigorously? 

Reviewer #1: Yes

4. Have the authors made all data underlying the findings in their manuscript fully available?

Reviewer #1: Yes

5. Is the manuscript presented in an intelligible fashion and written in standard English?

Reviewer #1: Yes

6. Review Comments to the Author

Reviewer #1: (No Response)

7. PLOS authors have the option to publish the peer review history of their article (what does this mean?). If published, this will include your full peer review and any attached files.

Reviewer #1: **Yes: **Juanwen Yu
